# LESION SEARCH WITH SELF-SUPERVISED LEARNING

**Kristin Qi**[1]  **Jiali Cheng**[2]  **Daniel Haehn**[1]

[1]University of Massachusetts Boston, Boston, USA    [2]University of Massachusetts Lowell, Lowell, USA

{qi,haehn}@mpsych.org   jiali_cheng@uml.edu

## ABSTRACT

Content-based image retrieval (CBIR) with self-supervised learning (SSL) accelerates clinicians' interpretation of similar images without manual annotations. We develop a CBIR from the contrastive learning SimCLR and incorporate a generalized-mean (GeM) pooling followed by L2 normalization to classify lesion types and retrieve similar images before clinicians' analysis. Results have shown improved performance. We additionally build an open-source application for image analysis and retrieval. The application is easy to integrate, relieving manual efforts and suggesting the potential to support clinicians' everyday activities.

## 1 INTRODUCTION

Radiologists' interpretation of lesions (nodules, tumor-like, and surface-like) requires careful analysis. Lesion images employ a wide variety of sizes, shapes (e.g., elliptical, irregular), colors (e.g., grayscale), and textures (e.g., convex, nodule, distortion) (Hofman & Hicks (2016)), which makes manual image retrieval or content-based image retrieval (CBIR) with annotations time-consuming and at risk of errors. Annotating lesions is expensive because of intra-class variations: the same lesion type may not be visually similar, while different lesion types may appear similarly once images are taken at similar stages of disease progression. In this work, we first develop the SSL method for lesion feature extraction that is prepared for image retrieval and classification tasks. The overall pipeline is shown in Figure 1. Second, we developed a web-based open-source DICOM (Digital Imaging and Communications in Medicine) application for Radiology image analysis and similar lesion image retrieval, shown in Figure 2. Easy Python installation can be found at https://github.com/openhcimed/flask_search.

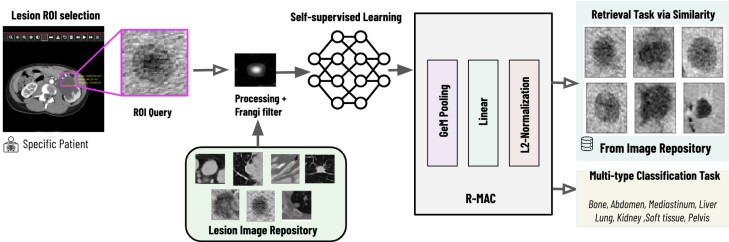

Figure 1: Pipeline of the front-end and back-end framework and downstream tasks.

## 2 APPLICATION: DICOM USER INTERFACE

The application comprises two components: UI front-end and CBIR back-end. **UI front-end** is an AngularJS-based viewer with toolbars. Users can drag and drop single or batch images for analysis, such as adjusting timeframes, zooming, color contrast, annotations, and viewing metadata. Annotations are in JSON format and can be saved locally or sent to the CBIR back-end SSL model.

**Back-end model.** First, we preprocessed regions of interest and applied the Hessian-based Frangi filter (Frangi et al. (1998); Kroon & Schrijver (2009); Longo et al. (2020)) to maximize lesions while minimizing noise. Details are in Appendix A. Second, the baseline started with a robust SimCLR (Chen et al. (2020)) with ResNet-18 (He et al. (2016)). To improve image retrieval, we used

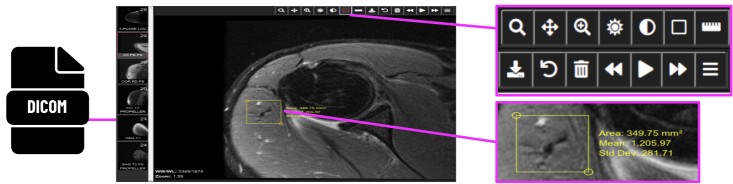

Figure 2: The front-end UI can load multiple series of DICOM files. It offers a variety of functionalities such as annotations, size measures, contrasts, navigating image stacks, etc.

a generalized-mean (GeM) pooling layer (Radenović et al. (2018)) followed by L2 normalization. We trained networks on the public DeepLesion dataset (Yan et al. (2018)) without patients' identifications, reliving data privacy concerns. Data details are in Appendix B. Finally, CBIR ranked candidate descriptors $C$ in ascending order according to their closest cosine distances to a query $\mathbf{Q}$. We chose top-9 candidates to cover enough similarity. Furthermore, experimental details are in Appendix C. Loss function and distance details are in Appendix D.

## 3 RESULTS

**Classification results.** We started comparisons between two baselines: SimCLR and VAE (Variational AutoEncoder), utilizing ResNet-18. Table 1 shows that either applying the Frangi filter during preprocessing or GeM pooling layer with L2-normalization results in improvements from baselines, but to a different extent. Frangi filter provides a much stronger improvement over SimCLR, yet combining it with the GeM approach results in the most effective feature extractor. However, using Frangi filter has **limitations**, as varying parameters may produce discrepancies within the dataset or in a different dataset. We observed benefits from these combinations, but it's apparent that these methods are off-the-shelf and are not compared with novel SSL networks. Furthermore, the performance on other medical datasets to solve more problems is unknown, which is left for future work.

Table 1: **Lesion type classification comparisons.** SimCLR or VAE with ReNet-18 were trained as baseline models. *Frangi*: filter in preprocessing, *+GeM*: GeM pooling approach.

| Base model | Frangi | +GeM | Accuracy (%) | F1-score (%) |
|------------|--------|------|--------------|--------------|
| SimCLR | | | 64.52 | 65.57 |
| SimCLR | ✓ | | 80.39 | **78.93** |
| SimCLR | | ✓ | 69.24 | 67.74 |
| SimCLR | ✓ | ✓ | **80.73** | 77.83 |
| VAE | | | 74.04 | 68.83 |
| VAE | ✓ | | 74.35 | **75.79** |
| VAE | | ✓ | **76.34** | 73.78 |
| VAE | ✓ | ✓ | 76.04 | 75.24 |

**CBIR results.** As shown in Appendix E Table 2, we assess the same patient (intra-patient), across different patients (inter-patient), and all patients, as per commonly used clinical evaluations. The standard retrieval metrics, mAP@10 and Precision@k, suggest that intra-patient retrieval precision is higher because of highly similar features within a single patient, while features are dissimilar between different patients. Furthermore, results indicate that the SimCLR model (contrastive-based) outperforms the VAE model (generative-based).

## 4 CONCLUSIONS

We present an open-source interactive application that facilitates lesion analysis and retrieval based on self-supervised learning (SSL). Developed from the contrastive learning SimCLR, with Frangi filter, GeM pooling and L2-normalization together improve lesion retrieval performance.

## URM Statement

The authors acknowledge that at least one key author of this work meets the URM criteria of ICLR 2023 Tiny Papers Track.

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

APPENDIX

## A  FRANGI FILTER FOR LESION DETECTION

Frangi filter function has two components and $\lambda$ represents eigenvalues (Frangi et al. (1998)): if either $\lambda_2 > 0$ or $\lambda_3 > 0$, $V_f(\lambda) = 0$. Otherwise:

$$V_f(\lambda) = (1 - e^{-\frac{R_A^2}{2\alpha^2}}) \cdot e^{-\frac{R_B^2}{2\beta^2}} \cdot (1 - e^{-\frac{s^2}{2\gamma^2}}), \tag{1}$$

where $R_A = \frac{|\lambda_2|}{|\lambda_3|}$, $R_B = \frac{|\lambda_1|}{\sqrt{|\lambda_2 \cdot \lambda_3|}}$, and $s = \sqrt{\lambda_1^2 + \lambda_2^2 + \lambda_3^2}$. The modification enhances contours by restricting $\lambda$ such that $|\lambda 1| \leq |\lambda_2| \leq |\lambda_3|$ at the scale "s". To suppress background noises that are not contours, we set $\lambda_3 = 0$ if $\lambda_3 > 0$, and change $\lambda_1$ and $\lambda_2$ into high eigenvalues to control $R_B$ close to 1, which differentiates blob-like and plate-like lesion structures from others. Threshold parameters are $\alpha = 1$, $\beta = 0.6$, and $\gamma = 0.0444$ to balance sensitivity differentiating blob-like and plate-like lesions from background noises. To capture lesions with various sizes, we modify the multiscale value, "s", from 1 to 9 with a step size of $0.2$. A few examples comparing original ROIs and responses after applying the new function are shown in Figure 3.

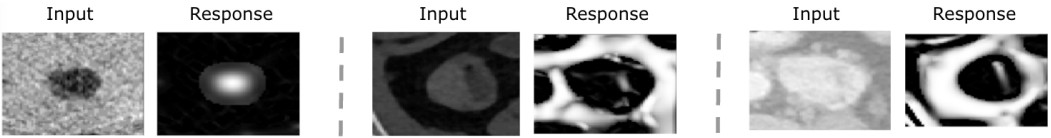

Figure 3: Sample responses after the modified Frangi filter for lesion contour detection.

## B  DATASET

**DeepLesion Dataset.** Previously, most algorithms considered one or a few types of lesions (Pham et al. (2018); Tariq et al. (2021); van Tulder & de Bruijne (2016)). Yan et al. developed the large and publicly available DeepLesion dataset from the NIH Clinical Center (Yan et al. (2018)) towards developing universal lesion detections. In clinical practice, DeepLesion dataset is widely accepted for monitoring cancer patients due to its recorded diameters in accordance with the Solid Tumor Response Evaluation Criteria (RECIST), which is one of the commonly used means of criteria. DeepLesion dataset consists of 33,688 PACS-bookmarked CT images from 10,825 studies of 4,477 unique patients (Yan et al. (2018)). It includes various lesions across the human body, such as lungs, lymph, liver, etc. Four cardinal directions (left, top, right, bottom) enclose each lesion in every CT slice as a bounding box to mark coarse annotations with labels. We crop regions of interest (ROIs) based on these bounding boxes instructed in the dataset to comprise excessive instances for model training.

## C  EXPERIMENTAL SETUP

**Data pre-processing.**   We preprocess DeepLesion by cropping bounding boxes, flips, color distortions, and Gaussian blur. We keep the image size as 64 x 64 and apply the Frangi filter.

**Implementation Details.**   We train models with Pytorch (Krizhevsky et al. (2012)) on a single Nvidia DGX-A100 GPU. We implement ResNet-18 (He et al. (2016)) in SimCLR, followed by two more convolutional layers and a GeM pooling layer with L2-normalization. We performed 1000 epochs for SSL pre-training with LARS (You et al. (2017)) optimizer of a $0.05$ learning rate, a $10^{-5}$ weight decay, and a cosine learning rate scheduler. For CBIR task, we compute contrastive loss with margin 0.8 and cosine distance from the sigmoid classification layer. Optimized with SGD, we fine-tune the model for 50 iterations with a learning rate of 0.01, momentum of 0.9, and tracker of cosine growth for the learning rate.

## D  CBIR DESCRIPTORS FOR THE MOST SIMILAR LESIONS.

We used the following loss function: (Gómez (2019))

$$L(a, p, n) = \frac{1}{2} max(0, m + d(a, p) - d(a, n)) \tag{2}$$

$a$ is an anchor, $p$ is a positive sample sharing the same lesion type with $a$. $n$ is a negative sample different from $a$'s type. $m$ is the margin determining the stretch for separating negative and positive samples. We use cosine distance similarity measure $d$ to imply how similar a retrieved candidate is to a given query:

$$d(i, j) = \frac{i \cdot j}{\|i\| \, \|j\|} \tag{3}$$

$i$ is a query and $j$ is a lesion candidate embedding.

## E  CBIR RESULTS

Table 2: **CBIR performance comparisons.** Evaluations on VAE or SimCLR approach: a) All-patients (all lesions are candidates); b) Same-Patient (lesions of the same patient are candidates); c) Cross-patient (lesions from different patients are candidates).

| Evaluation setting | Method | mAP@10 | Precision@1 | Precision@10 |
|---|---|---|---|---|
| All-patients | VAE | 0.688 | 0.699 | 0.581 |
| | SimCLR | **0.729** | **0.734** | **0.643** |
| Same-patient | VAE | 0.707 | 0.746 | 0.630 |
| | SimCLR | **0.713** | **0.757** | **0.633** |
| Cross-patient | VAE | 0.378 | 0.408 | 0.370 |
| | SimCLR | **0.465** | **0.503** | **0.458** |

