# OpenReview forum: "Lesion Search with Self-supervised Learning"
_ICLR.cc/2023/TinyPapers — Submitted to Tiny Papers @ ICLR 2023_

### Official Review · Reviewer_MQFp · 2023-03-29

**Confidence:** 3

**Summary Of Contributions:**

The paper proposes a self-supervised pipeline to provide lesions analysis, together with an user interface to ease the medics work.

**Rating:**

Great Start (GS): a submission which meets some of the reviewing criteria but has room for improvement

**Strengths And Weaknesses:**

STRENGTHS
=======
The problem looks relevant and sensitive. I like the idea of introducing an interactive tool to ease the work of users, which might lead to a good impact by removing some significant technological gatekeepers.

WEAKNESSES
======
1) The paper does not respect the Tiny Papers format since it exceeds two pages.
2) The paper is clear and nice from an applicative perspective, but it misses some discussion about insights into the proposed method, future works and limitations. I am also quite confused by the bold numbers of Table 1, which make me unclear on how to compare different results. I am probably missing something, but how is it true that the method outperforms competitors, given that Yan et al. perform slightly better? Why is F1 score not reported for some of the competitors?

**Suggested Changes:**

To strengthen the submission, my suggestions are:
1) Rework the paper to make it fit two pages (probably moving one figure to the appendix should be enough)
2) Provide a better discussion on the main insight of the method: what would be promising for future works? What are the main limitations of the current approach? What is the main take-home message for the researchers?
3) Clarify the results in Table 2.

Minor:
- Typo: "... self-supervised learning, It simplifies and improves clinicians..."; the comma should be a period.
- Probably, a discussion about security\privacy incorporated in the tool would be good, given the sensitivity of the considered data.
- I assume the code will be released, but it probably would be good to mention this in the Introduction among the contributions.

---

### Official Review · Reviewer_53kf · 2023-04-02

**Confidence:** 3

**Summary Of Contributions:**

The paper proposes a self-supervised method for lesion classification and retrieval. The method uses simCLR with a ResNet18 backbone, with Frangi filter preprocessing.

**Rating:**

Great Start (GS): a submission which meets some of the reviewing criteria but has room for improvement

**Strengths And Weaknesses:**

Strengths:
- The paper clearly describes the problem and motivation. The task sounds important and the developed tool can be helpful to radiologists.
- Use of Frangi filters as a preprocessing step shows notable improvements in the classification accuracy.
- The experimental details are provided sufficiently.

Weaknesses:
- The main focus of the paper is on CBIR. However, the results for CBIR are not compared to any baselines, and more discussions and comparisons are required for this part.
- The baselines for the classification task (Random Forest and Logistic Regression) are too weak compared to the proposed method which uses ResNet18.
- An ablation for the GeM pooling block is helpful to show the impacts, similar to what has been done for the Frangi filters.
- The paper does not follow the formatting requirement and is longer than the page limit.


**Suggested Changes:**

Please see the weaknesses.

---

### Comment · Area_Chair_pJ1z · 2023-06-03
**Archival**

This work meets the threshold for archival, contents the URM statement and is deanonymized

---

### Meta-Review · Area_Chair_pJ1z · 2023-04-06

**Recommendation:** Invite to revise
**Confidence:** 5

**Metareview:**

This paper proposes to use SimCLR and ResNet-18 for the purpose of  lesion classification and retrieval. Together with an UI, the goal of this work is to ease the medical work.
Strengthes noted by the reviewers are that this paper could have the potential make a good impact on the medical domain, and the methods have shown performance improvements.
Yet several concerns remain, the performance is not compared with other baselines, more ablations are needed, and that the formatting of the paper exceeds two pages.


**Summary:**

A lesion classification and retrieval method is proposed using SimCLR and ResNet-18, reviewers agree that this paper could make impact, yet several concerns about the details and experiments remain. More importantly, the format of the paper violates the formatting rules.

**Reason For Not Giving A Higher Recommendation:**

The concerns about the experiments and details, and the formatting issues.


**Reason For Not Giving A Lower Recommendation:**

N/A

---

### Decision · Program_Chairs · 2023-04-07

**Decision:**

Revision accepted; invite to archive

**Comment:**

Authors are asked to please revise their paper to meet the formatting requirements and address reviewers' feedback.

---

> ### Author Response · Authors · 2023-06-01
> **Summarize the revision to address feedback**
>
> We wish to opt-in for archival, for sure.
>
> We are very grateful for the valuable feedback. All comments have been incorporated into the revision, along with more research and experiments. To summarize, we follow the page limit and discuss insights, take-home messages, limitations, sensitive data concerns, and future work. The discussions are based on a more comprehensive ablation study and a better presentation of result tables, as suggested. Adding the code link helps introduce our work.